# Genome–Wide Identification and Functional Characterization of Auxin Response Factor (ARF) Genes in Eggplant

**DOI:** 10.3390/ijms23116219

**Published:** 2022-06-01

**Authors:** Jing Chen, Shijie Wang, Fengling Wu, Min Wei, Jing Li, Fengjuan Yang

**Affiliations:** 1State Key Laboratory of Crop Biology, College of Horticulture Science and Engineering, Shandong Agricultural University, Tai’an 271018, China; qq1023186328@163.com (J.C.); shijie198811@163.com (S.W.); 18864805330@163.com (F.W.); minwei@sdau.edu.cn (M.W.); 2Scientific Observing and Experimental Station of Facility Agricultural Engineering (Huang–Huai–Hai Region), Ministry of Agriculture and Rural Affairs, Tai’an 271018, China; 3Shandong Collaborative Innovation Center for Fruit and Vegetable Production with High Quality and Efficiency, Tai’an 271018, China; 4Key Laboratory of Biology and Genetic Improvement of Horticultural Crop (Huang–Huai Region), Ministry of Agriculture and Rural Affairs, Tai’an 271018, China

**Keywords:** genome–wide, auxin response factor genes, eggplant, auxin signaling, 2,4–D, salt stress

## Abstract

Auxin response factors (ARFs) are important plant transcription factors that are differentially expressed in response to auxin and various abiotic stresses. ARFs play important roles in mediating plant growth and stress responses; however, these factors have not been studied in eggplants. In this study, genome–wide identification and the functional analysis of the ARF gene family in eggplants (*Solanum melongena* L.) were performed. A total of 20 *ARF* (*SmARF*) genes were identified and phylogenetically classified into three groups. Our analysis revealed four functional domains and 10 motifs in these proteins. Subcellular localization showed that the SmARFs localized in the nucleus. To investigate the biological functions of the *SmARFs* under 2,4–D and salt stress treatments, quantitative real–time RT–PCR (qRT–PCR) was conducted. Most *SmARF* genes exhibited changes in expression in response to 2,4–D treatments in the flowers, especially *SmARF4* and *7B*. All *SmARF* genes quickly responded to salt stress, except *SmARF17* and *19* in leaves, *SmARF1A* and *7B* in roots, and *SmARF2A*, *SmARF7B*, and *SmARF16B* in stems. These results helped to elucidate the role of ARFs in auxin signaling under 2,4–D and salt stress in eggplants.

## 1. Introduction

Eggplant (*Solanum melongena* L.) is an excellent source of fiber, vitamins, minerals, and antioxidant polyphenols, making it a popular food and an economically important vegetable crop species that is cultivated worldwide [1,2]. This plant species originates in Africa and is widely cultivated in Asia, Africa, Europe, and the Near East [3]. China is the largest producer of eggplant, accounting for 68% of global production in 2020 [4]. Flower abscission is a serious constraint to the yield and quality of the plant [5,6]. During planting, 2,4–D (2,4–dichlorophenoxyacetic acid), a synthetic auxin plant growth regulator, can be sprayed on flower buds to mitigate flower abscission and increase fruit set [7].

Auxin is a plant hormone that is strongly involved in several biological processes, including organ formation and development [8,9,10,11,12]. Two transcription factor families are associated with the auxin signal transduction pathway: the auxin response factor (ARF) and auxin/indole acetic acid (Aux/IAA) [13]. ARFs have a modular domain structure with three major domains. The N–terminal B3–like DNA–binding domain (DBD) is highly conserved and binds to auxin response elements (AuxREs) in the promoters of auxin–responsive genes [13,14]. The middle region activates or inhibits target genes, depending on their sequence [13,14]. The C–terminal dimerization domain (CTD) contains the motifs III and IV, which are also found in the Aux/IAA family.

The functions of ARFs have been studied in *Arabidopsis thaliana*, tomato (*Solanum lycopersicum*), papaya (*Carica papaya* L.), sugar beet, peach, and other plant species [13,15,16,17,18,19,20,21,22,23,24,25]. *AtARF 2*, *3*, *5*, *6*, *7*, *8*, *17*, and *19* regulate apical bud formation, pollen wall synthesis, vascular bundle development, hypocotyl tropic movement, and adventitious root formation [26,27,28,29,30]. For example, the phenotypes of *arf2* T–DNA insertion mutations indicated that *ARF2* regulates leaf senescence [31] and floral organ abscission [32]. *SlARF6A*, *8A*, *8B,* and *24* regulate leaf shape development [33]. *SlARF 3*, *4*, *5* act in tomato development and can stimulate the formation of epidermal cells and trichomes, improve plant resistance to water deficit, and regulate fruit set and development [34,35,36]. However, the biological functions of the *SmARF* gene family members in response to auxin in eggplants are unknown.

Soil salinization is one of the most crucial abiotic stresses that affects the growth of plant leaves, roots, and stems [37]. Various *ARF* genes act in response to salinity stress in some plants, such as sugar beet (*Beta vulgaris* L.), tomato (*Solanum lycopersicum*), rice (*Oryza sativa*), sweet potato, and *Tamarix chinensis* [19,38,39,40,41,42,43]. It has been reported that many *SlARFs* were differentially expressed in tomato leaves and roots under salt stress conditions, and all *SlARF* genes were down–regulated in leaves except *SlARF1*, *SlARF4*, and *SlARF19*, whose expression was significantly induced [38]. *SlARF8A* and *SlARF10A* were significantly up–regulated by salt and drought stresses [38], while the down–regulation and loss function of *SlARF4* increased tomato tolerance to salinity and drought stress [39]. In rice, both *OsARF11* and *OsARF15* showed differential expression under salt stress conditions [40]. Most *BvARF* genes in sugar beet were up–regulated or down–regulated to varying degrees by salt stress, and with significant changes in nine genes [19]. In *Tamarix chinensis,* the expression profiling of *TcARFs* showed that only *TcARF6* responds to salt stress [42].These reports suggest that ARFs respond to salt stress and participate in the crosstalk between auxin and salt stress signaling.

The newest eggplant genome [44] obtained from the scaffold of v3.0 using 3D chromosome conformation capture (Hi–C) information was used to identify the members of SmARF family using known SlARF protein sequences from tomato. We found 20 potential *ARF* (*SmARF*) genes and investigated the phylogenetics, gene structure, conserved motifs, and putative cis–acting regulatory elements (CAREs). We observed the subcellular location of 10 SmARFs, and gene expression patterns in various tissues and in response to hormones and salt stress were also determined. This is the first identification and characterization of *SmARFs* in eggplant, and the results should facilitate future work to explore the involvement of *SmARFs* in auxin–mediated responses, as well as in salt stress–tolerant eggplant varieties.

## 2. Results

### 2.1. Identification and Sequence Analysis of ARF Genes in Eggplant

Full–length putative ARF protein sequences were obtained by querying tomato ARFs [17] against the eggplant genome reported by Barchi et al. (2021) [44] using BLASTP. A total of 20 candidate genes were identified, and the presence of SmARF domains was evaluated via the comparison of these sequences to other ARFs in the Pfam databases. To achieve the consistent nomenclature of ARF genes across species, eggplant members of this gene family were named based on their phylogenetic relationship, and by the numbering of the closest tomato homologs. All SmARF proteins were found to contain an ARF domain (Table 1), as revealed through the Pfam analysis tool http://pfam.sanger.ac.uk/ (accessed on 15 December 2021). Of the identified genes, SmARF19 and SmARF16A proteins contain the longest (1423 aa) and shortest (379 aa) amino acid sequences, with a mean of 824.85 aa. The molecular weight of the SmARF proteins ranged from 43.09 (SmARF6A) to 158.50 (SmARF19) kDa, with a mean of 90.46 kDa. The pI varied from 5.44 (SmARF8A) to 8.06 (SmARF6A), with a mean of 6.41. The localization of all SmARF proteins were predicted in the nucleus (Table 1).

An alignment of eggplant ARFs was made against tomato homolog ARFs protein sequences, with the three highly conserved domains (B3, auxin_resp, and Aux/IAA–binding domain) aligned in multiple (Figure 1). Of the 20 sequences, all but SmARF19 included a typical DBD domain (B3), with a high conservation of the N–terminal DBDs in these proteins. All of the putative SmARFs contained an auxin_resp domain. However, only nine SmARF proteins (SmARF1A, 1B, 2A, 2B, 5, 7A, 7B, 18, and 19) contained an Aux/IAA–binding domain (Figure 1).

### 2.2. Phylogenetic Tree Analysis and Grouping of Auxin Response Factor (ARF) Genes in Eggplant

To analyze the evolutionary relationship between candidate SmARF genes, a phylogenetic tree was constructed using ARF protein sequences from eggplant (20), *Arabidopsis* (23) [13], and tomato (22) [17] (Figure 2). In total, the 67 sequences included in the phylogenetic tree were grouped into three major clades (I, II, and III; Figure 2). Group I included five SmARF proteins, five SlARF proteins, and three AtARF proteins. Group II had seven SmARFs, nine SlARFs, and five AtARFs. Group III was divided into four subgroups, designated Group IIIa, b, c, and d. Group IIIa included two SmARFs, two SlARFs, and two AtARFs. Group IIIb contained four SmARFs, three SlARFs, and two AtARFs. Group IIIc had two SmARFs, four SlARFs, and two AtARFs. Subclass IIId contained no SmARF or SlARF sequences, but included nine AtARFs (Figure 2). A total of 20 sister pairs were formed in the combined phylogenetic tree, with sixteen SmARF–SlARF pairs, one SmARF–SmARF pair, one SlARF–SlARF pair, and two AtARF–AtARF pairs. Groups I, II, IIIa, IIIb, IIIc, and IIId all included at least one member from the SmARFs, SlARFs, and AtARFs, but group IIId contained only AtARFs (Figure 2).

### 2.3. Motif Analysis

The structural features of the SmARF proteins were analyzed using MEME, and 10 conserved motifs were identified (Figure 3). Motifs 2, 3, 4, and 6 were detected in most ARF proteins (Figure 3). All of the motifs common or specific to particular groups are represented in the phylogenetic tree (Figure 3). Motif 5 was not found in any genes in group I. In group II, all of the genes contained motifs 1–10, except SmARF6A, which lacked motif 5, 8, and 10, and SmARF8A, which lacked motif 1 and motif 5. Group IIIa, b, and c also contained all motifs, in addition to SmARF3, which lacked motifs 5 and 10, SmARF18 which lacked motif 9, and SmARF24, which lacked motifs 5, 9, and 10.

### 2.4. Prediction of Cis–Acting Elements in Promoters 

Promoters control gene expression, and transcription factors can regulate gene expression by binding to cis–acting elements. Various cis–acting elements either induce or inhibit gene expression in response to different biotic or abiotic stress signals. We predicted and analyzed potential cis–acting elements in the promoters of all the identified *SmARFs* using the PlantCARE tool. Several well–characterized elements were identified in the promoter regions of *SmARFs*, including light–responsive, low–temperature responsive, gibberellin–responsive, methyl jasmonate (MeJA)–responsive, abscisic acid–responsive, defense–responsive, stress–responsive, salicylic acid–responsive, and wound–responsive elements, in addition to MYB–binding sites (Figure 4). Multiple elements were present in each promoter. The identified light–responsive elements included TCT–motif, LAMP–element, I–box, GT1–motif, G–box, GATT–motif, GA–motif, Box 4, ATCT–motif, and AE–box.

### 2.5. Subcellular Localization of SmARFs 

Ten SmARFs (SmARF1A, SmARF6A, SmARF2B, SmARF1B, SmARF18, SmARF8A, SmARF10B, SmARF24, SmARF16A, and SmARF10A) from each of the three clades were selected for the analysis of subcellular localization. To perform this, SmARF:GFP dual–expression vectors were constructed for each *SmARF*, and a GFP construct was used as a positive control. Fluorescence microscopy revealed that the GFP signals of all 10 fusion proteins were detected in the nucleus (Figure 5).

### 2.6. Expression Patterns of SmARF Genes in Eggplant

To explore the function of the SmARFs throughout the plant, the expression patterns of the *SmARF* genes were determined in eggplant leaves, roots, and stems using qPCR (Figure 6). Some *SmARFs* were differentially expressed in leaves (*SmARF1A*, *1B*, *2B*, and *19*), roots (*SmARF5*, *6A*, *7B*, and *24*), and stems (*SmARF3*, *4*, *6B*, *7A*, *8A*, *10A*, *16B*, *17*, *18*).

### 2.7. Expression Analysis of SmARF Genes in Response to 2,4–D

ARFs are sensitive to auxin, and they are strong regulators of auxin signal transduction pathways. A synthetic growth regulator, 2,4–D, is usually sprayed on the flower stalk to reduce flower abscission and increase fruit set. To determine how *ARFs* are involved in the response to auxin, the 3 month–old eggplant flower buds were treated before blooming with 2,4–D for 0 h, 2 h, 4 h, and 6 h and then gene expression was analyzed using qPCR (Figure 7).

A two–hour 2,4–D treatment increased the expression of *SmARF2B*, *SmARF6A*, *SmARF6B*, *SmARF7A*, *SmARF10B*, *SmARF16A*, *SmARF19*, and *SmARF24* by more than 2–fold, and decreased the expression of *SmARF1B*, *SmARF2A*, *SmARF**5*, *SmARF8A*, *SmARF16B*, and *SmARF18* by more than 2–fold. However, after four hours of exposure, the expression levels of *SmARF**3*, *SmARF6A*, *SmARF6B*, *SmARF7A*, *SmARF10B*, *SmARF16A*, *SmARF19*, and *SmARF24* decreased to lower levels than without treatment. A six–hour treatment increased the expression of *SmARF**3*, and *SmARF17*.

### 2.8. Expression Analysis of SmARF Genes in Response to Salt Stress

During cultivation, soil secondary salinization is one of the main stresses that can limit the growth of eggplants. Here, we treated eggplant seedlings with 250 mm NaCl for 0 h, 2 h, and 24 h, and then analyzed the expression of the 20 *SmARF* genes using qRT–PCR (Figure 8). 

All of the *SmARF**s* exhibited rapid responses to salt stress in leaves except *SmARF**4/5/7B/**17**/**19*, in roots except *SmARF1A**/5*/*7B*, and in stems except *SmARF**5*. A two–hour salt treatment increased the expression of seven genes (*SmARF**2A*, *SmARF**10A*, *SmARF**10B*, *SmARF**16A*, *SmARF1**6B*, *SmARF1**8*, and *SmARF16*) in leaves, three genes (*SmARF**1B*, *SmARF**7A*, and *SmARF**10B*) in roots, and 15 genes (*SmARF**1A*, *SmARF**1B*, *SmARF**2B*, *SmARF**3*, *SmARF**4*, *SmARF**6A*, *SmARF**7A*, *SmARF**7B*, *SmARF**8A*, *SmARF1**0B*, *SmARF**16A*, *SmARF1**7*, *SmARF18*, *SmARF19*, and *SmARF**24*) in stems, suggesting an involvement of the ARF genes in salt stress responses in eggplant.

## 3. Discussion

Multiple ARF transcription factors have been reported for *Arabidopsis thaliana* [13], tomato (*Solanum lycopersicum*) [17], papaya (*Carica papaya* L.) [18], longan (*Dimocarpus longan* L.) [23], apple (*Malus domestica*) [25], and rice (*Oryza sativa*) [45], with 23, 22, 11, 17, 31, and 25 genes, respectively. In this study, we identified and characterized 20 SmARF transcription factors in eggplant. They were unevenly distributed on 12 chromosomes and have highly similar domains. Each ARF generally consists of three conservative domains, B3, auxin_resp, and Aux/IAA–binding [46]. Among the 20 *SmARFs*, all but SmARF19 included a typical DBD domain (B3). The lack of a B3 domain suggests that the protein encoded by this gene is not able to recognize and bind to the auxin response element in the promoter sequences of target genes [45]. As transcription factors, ARF proteins generally act in the nucleus. Consistent with the subcellular location of ARF in other species, all SmARF members are predicted to be in the nucleus, consistent with their function as transcription factors [17,47]. Our results with fluorescent reporters confirmed nuclear localization for 10 of these proteins (Figure 5).

A phylogenetic tree was constructed to analyze relationships between ARF family genes in eggplant, *A. thaliana*, and tomato (Figure 2). The results showed that most sister gene pairs with high bootstrap values (≥99%) were identified between eggplant and tomato. The absence of any *SmARF* gene in subclass IIId suggested that the nine duplicated *AtARF* genes of this group were originally derived from a single *AtARF* gene. This result was consistent with the tomato ARFs [25], suggesting that ARFs in eggplant were highly homologous to those in tomato. Conserved motifs for transcription factors correlate with protein interactions, transcriptional activity, and DNA–binding [24]. Ten conserved motifs were identified in this study (Figure 2). All the motifs common or specific to particular groups were represented in the phylogenetic tree (Figure 3). Although the number of members in each phylogenetic group varied, there was a strong conservation of the patterns of motifs within a group. The comparison of sequences of novel functional domains/motifs across multiple orthologous proteins is an approach that is widely used to predict protein functions based on evolutionary conservation. Although most motifs in SmARFs are conserved, other motifs may be associated with novel functions in plants and should be further investigated. Defense–responsive, stress–responsive, light–responsive, low temperature–responsive, drought–responsive, gibberellin–responsive, MeJA–responsive, abscisic acid–responsive, salicylic acid–responsive, and wound–responsive elements, as well as MYB–binding sites, were identified upstream of *SmARF* genes (Figure 4).

To identify the functions of *SmARFs* in eggplant, we analyzed their expression in leaves, roots, and stems. The data indicated that the expression of *SmARFs* was ubiquitous in all tissues. Among these genes, *SmARF2B* was significantly highly expressed in leaves. Similarly, the expressions of its tomato homolog, *SlARF2B,* were also significantly higher in leaves [25]. In *Arabidopsis thaliana, AtARF2* could regulate leaf senescence [31]. Thus, we speculated that *SmARF2B* might play a similar role in eggplant leaves. However not all cases were the same. For example, *SmARF10A* was significantly high expressed in stems, while *SlARF10A* has been reported to be involved in the spatial restriction of the auxin response that drive leaf blade outgrowth in tomato [38]. Further work will investigate the function of *SmARFs* on growth and development of eggplants.

As transcription factor, *ARFs* participate in signal pathways related to auxin response and regulate the flower–to–fruit transition [38,39,40,41,42,43,48]. The growth regulator 2,4–D was used to mitigate flower abscission and increase fruit set. In this study, the expression of *SmARF2B* was significantly increased in flower buds after 2,4–D treatment (Figure 7). The expression of its tomato homolog, *SlARF2,* was clearly responded to auxin. In addition, *SlARF2* could regulate lateral root formation and flower organ senescence [49]. The silenced *AtARF2/3/4* line leads to abnormal morphology of pollen grains [32]. The expression of *SmARF7A* was significantly increased after 2,4–D treatment for 2 h. The highly expression level of *SlARF7* in placental tissues of the mature flower could activate the auxin response, attenuating genes that might repress the auxin response and prevent fruit set [50]. We also found that *SmARF10A* was significantly increased after 2,4–D treatment for 4 h. It has been proven that *SlARF10A*, the target of sly–miR160, regulated auxin–mediated ovary patterning, as well as floral organ abscission and lateral organ lamina outgrowth [51]. Thus, we speculated that *SmARFs* can exhibit many different functions in different plant species.

Transcription factors (TFs) coordinate gene expression by activating or inhibiting transcription in response to various abiotic stress signals [30,38,39,40,41,42,44,45,46,47,48,49,50,51,52]. In this study, the expression patterns of *SmARF* genes in response to NaCl treatment indicated that they might have crucial roles in eggplant leaves, roots, and stems (Figure 8). For example, the expression of *SmARF2A* was significantly decreased in 2 h and increased 24 h after salt treatment. Previous studies have reported that the up–regulation of *SlARF2* results in various asexual reproduction growth phenotypes, including increased lateral root formation [49]. The inhibition of *AtARF2*, *AtARF3*, and *AtARF4* expression via tasiRNAs may release the repression of *Arabidopsis* lateral root growth [48]. In addition, *SmARF7A* was significantly repressed or induced, respectively, in leaves, roots, and stems under salt stress, which indicates that *SmARF7A* might be involved in the stress response within eggplant. Similar expression profiles werealso found in *SlARF7A*, which is phylogenetically close to *SmARF7A* from tomato [38]. It had shown that *At**ARF7* is involved in the control of lateral root formation in *Arabidopsis* [30]. The other ARF transcription factors also played an essential role in the response to phytohormones and abiotic stress. For example, the *IbARF5* gene from sweet potato (*Ipomoea batatas*) could increase its tolerance to salt and drought stress in transgenic *Arabidopsis* [41]. In *Tamarix chinensis*, *TcARF6* was rapidly expressed in response to salt stress, but it was significantly downregulated specifically in the roots [42].

## 4. Materials and Methods

### 4.1. Identification of ARFs in the Eggplant Genome

*SmARF* gene sequences were obtained from the Eggplant Genome database https://solgenomics.net/organism/Solanum_melongena/genome (accessed on 18 December 2021) [44], the *S. lycopersicum* database https://solgenomics.net/organism/Solanum_lycopersicum/genome (accessed on 18 December 2021) [53], and the *Arabidopsis thaliana* database https://www.arabidopsis.org (accessed on 18 December 2021) [54].

To identify ARFs and to remove redundant sequences, “auxin response factor” was input as keywords to search the database and genome websites. The number of amino acids, molecular weight, and theoretical isoelectric point (pI) of putative ARF sequences were calculated using EXPASy https://web.expasy.org/protparam (accessed on 25 December 2021). Subcellular localization was predicted using the BUSCA tool.

### 4.2. Phylogenetic Analysis of ARF Genes

Full–length protein sequences were used for phylogenetic analysis. The protein sequences were aligned using ClustalX software with default parameters [55]. Phylogenetic trees were generated using MEGA7 software, with a bootstrap test of 1000 replicates [56]. The final tree was viewed and modified using Interactive Tree Of Life (iTOL) software https://itol.embl.de/personal_page.cgi (accessed on 3 January 2022) [57].

### 4.3. Structural Characterization

The subcellular location and intron numbers of ARFs were obtained from genome databases. Conserved motifs were identified using MEME software version 5.0.5 http://meme–suite.org/tools/meme (accessed on 5 January 2022) with the following parameters: any number of repetitions, a maximum of 10 misfits, and an optimum motif width of 6–200 amino acid residues [58]. The upstream sequences (2 kb) of *SmARFs* were also retrieved from the genome database, and regulatory elements were identified by analysis using the PlantCARE database http://bioinformatics.psb.ugent.be/webtools/plantcare/html (accessed on 7 January 2022) [59]. Next, the structure and composition of *Sm**ARF* genes were investigated using TBtools software [60]. The conserved motifs senquences were listed in Appendix A and the Cis-acting elements on promoters of SmARFs were listed in Appendix A.

### 4.4. Subcellular Localization

The complete coding sequences of 10 *SmARF*s (*SmARF1A*, *SmARF6A*, *SmARF2B*, *SmARF1B*, *SmARF18*, *SmARF8A*, *SmARF10B*, *SmARF24*, *SmARF16A*, and *SmARF10A*) were obtained from the Eggplant Genome database [44]. HindIII and BamHI enzymes were used to linearize the CaMV35S promoter vector with GFP. Sequences of 10 *SmARF*s were combined into the vector using In–Fusion Snap Assembly cloning kits (Takara, Dalian, China) according to the manual. The fusion construct was transferred into *Agrobacterium tumefaciens* strain GV3101, and subcellular localization assays were performed as previously reported [61]. The protein localization was determined under 20× confocal microscopy. The primer sequences were listed in Appendix A.

### 4.5. Growth Conditions and Treatments 

Seeds of the high generation inbred line No. 108 [62] were sterilized for 10 min in 50% sodium hypochlorite, rinsed four times with sterile distilled water, and sown in pots containing peat. Seeds were grown at 25 ± 2 °C under a light regimen of 16 h light and 8 h dark at a temperature of 16 ± 2 °C for 40 days. The plants were then transferred to a greenhouse and grown under the same temperature and photoperiod.

For chemical treatment, the herbicide 2,4–dichlorophenoxyacetic acid (2,4–D; BDH Chemical) was dissolved in water according to the manual, and the pH was adjusted to 7.0. Flower buds with no obvious pests, diseases, or mechanical damage were soaked in 1 mL of 1.0 mM 2,4–D for 0, 2, 4, and 6 h. 

For salt stress treatment, plants with three true leaves were treated with 250 mM NaCl solution (salt stress group). Leaves, stems, and roots samples were harvested after 0, 2, and 24 h treatment.

### 4.6. RNA Extraction and qRT–PCR Analysis

All samples were frozen in liquid nitrogen and stored at −80 °C for RNA extraction and other analyses. Total RNA was extracted from flower buds, leaves, roots, and stems using the RNA Isolation Kit (Ambion, Thermo Fisher Scientific, Waltham, MA, USA) following the manufacturer’s protocol, and stored at −80 °C. RNA integrity was evaluated using the Agilent 2100 Bioanalyzer (Agilent Technologies, Santa Clara, CA, USA).

First–strand cDNA was synthesized from 1 μg of total RNA using a Prime Script RT Reagent Kit (Takara, Dalian, China). PCR reactions were performed using the ABI 7500 Fast Real–Time PCR system (Applied Biosystems, Foster City, California, USA) and the QuantiFast SYBR Green PCR Kit (Qiagen, Duesseldorf, Germany). The amplification parameters were 95 °C for 5 min, followed by 45 cycles at 95 °C for 10 s, 60 °C for 10 s, and 72 °C for 10 s. The mRNA expression levels were normalized to the level of *SmActin* expression using the 2^−ΔΔCt^ method [49]. Each experiment included three biological replicates. The primer sequences are listed in Appendix A.

### 4.7. Statistical Analysis

In all the presented figures, error bars indicated standard deviation. SPSS 23.0 (SPSS, Inc., Chicago, IL, USA) was used to assess statistical significance using one–way ANOVA and Duncan’s New Multiple Range test (*p* < 0.05).

## 5. Conclusions

This is the first study of the evolution, expression profiles, and putative functions of ARF genes in eggplants. In the current study, 20 SmARF members were identified from the eggplant genome, and divided into three clades. Their gene structures were similar, and most members have a conserved ARF domain. Fluorescent reporters confirmed nuclear localization for 10 of these proteins. These members were highly expressed in leaves (*SmARF1A*, *1B*, *2B*, and *19*), roots (*SmARF5*, *6A*, *7B*, and *24*), and stems (*SmARF3*, *4*, *6B*, *7A*, *8A*, *10A*, *16B*, *17*, and *18*). These genes may play important roles in regulating eggplant development. The similar expression profiles of *SmARF2B/2A/7A/10A* to *SlARF2/7/10A* when under 2,4–D or salt treatment showed that these genes may play a similar role in regulating root formation and flower organ senescence, fruit set and lateral root formation, and floral organ abscission and lateral organ lamina outgrowth. These data will help to improve genotype selection, gene function analysis, and the improvement of agronomic traits in eggplants.

## Figures and Tables

**Figure 1 ijms-23-06219-f001:**
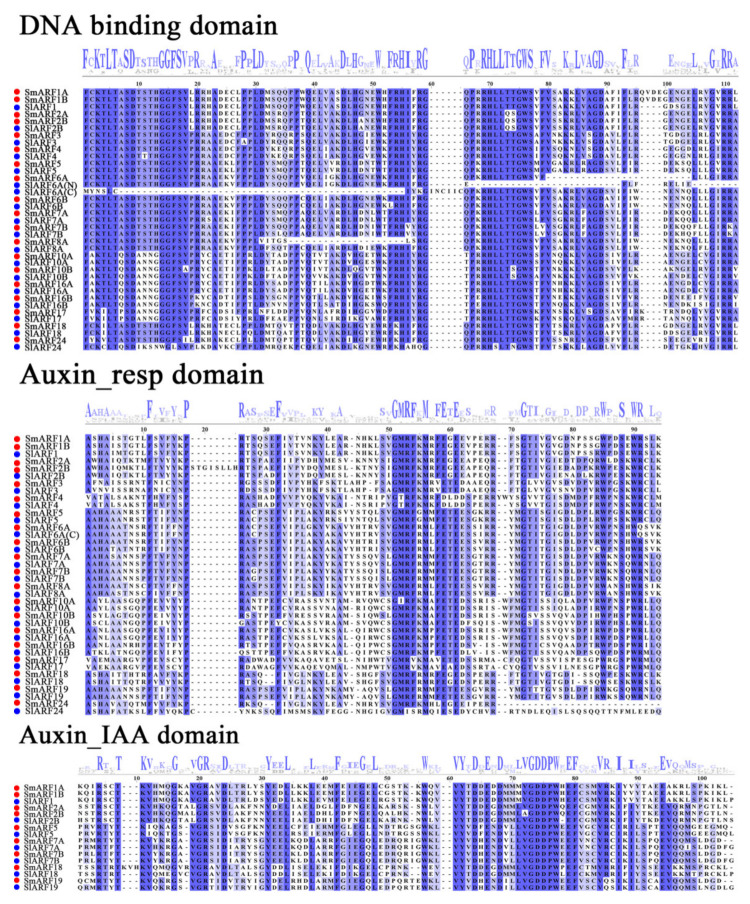
Multiple alignment profiles of the B3 (DNA binding domain), auxin_resp, and CTD (Aux/IAA–binding) domains of the SmARF proteins obtained with the Jalview program. Red and blue codes were used to distinguish the eggplant and tomato sequences. All sequences show high levels of amino acid conservation in blue. The logo at the top shows amino acids, and large letters represent greater conservation.

**Figure 2 ijms-23-06219-f002:**
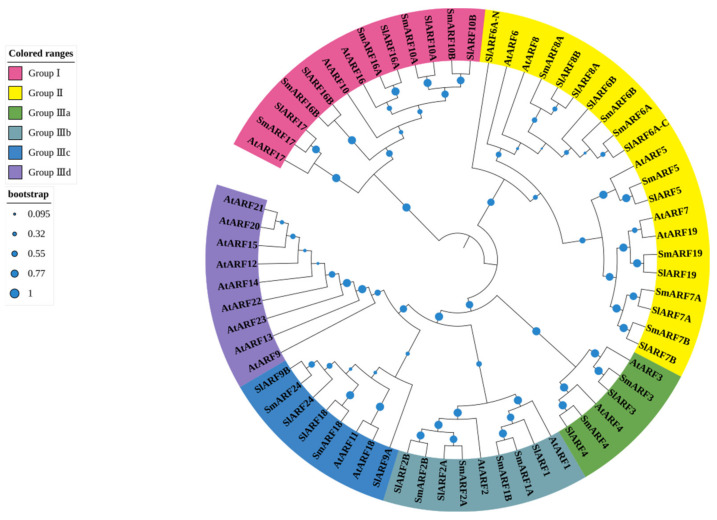
Phylogenetic analysis of genes encoding auxin response factors (ARFs) in *Arabidopsis*, eggplant, and tomato. A phylogenetic tree was constructed using MEGA 7 software through the neighbor–joining method, with 1000 bootstrap replicates using ARF protein sequences from eggplant (20), *Arabidopsis* (23), and tomato (22).

**Figure 3 ijms-23-06219-f003:**
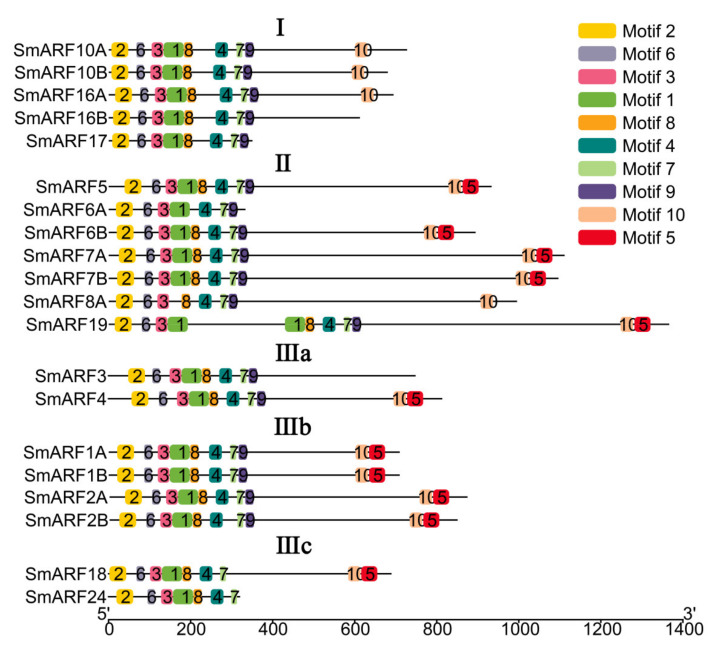
Schematic representation of conserved motifs in auxin response factor proteins in eggplant (*Solanum melongena* L.). The horizontal scale indicates protein length (number of amino acids).

**Figure 4 ijms-23-06219-f004:**
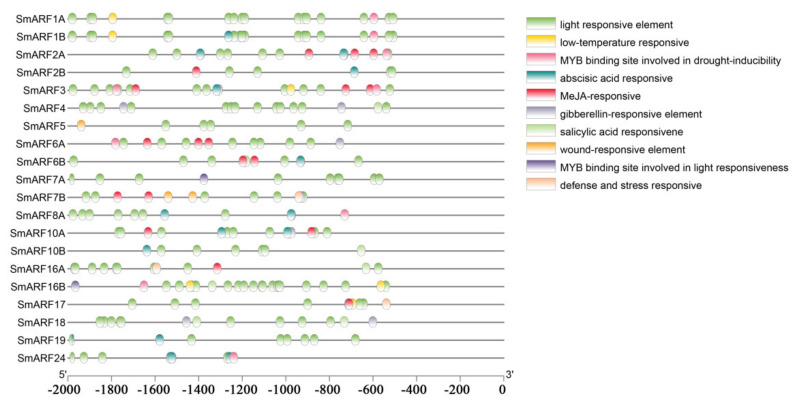
Cis–acting regulatory elements upstream of auxin response factor genes in eggplant (*Solanum melongena*). The presence of cis–acting elements in a 2000–bp sequence upstream of the ATG start codons was assessed using PlantCARE. The horizontal scale indicates promoter length.

**Figure 5 ijms-23-06219-f005:**
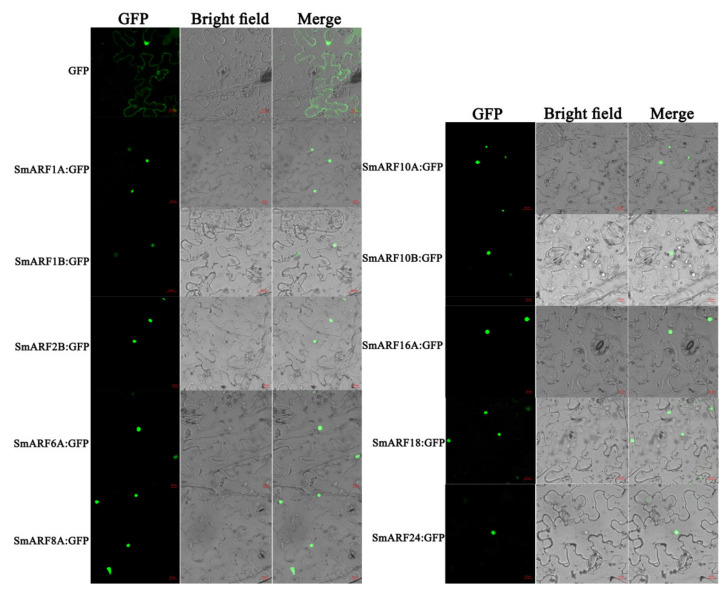
Subcellular localization of SmARFs (bars = 20 μm). SmARF:GFP fusion proteins were transiently expressed in tobacco leaves, and their localization was determined using confocal microscopy. The green dots correspond to the nucleus.

**Figure 6 ijms-23-06219-f006:**
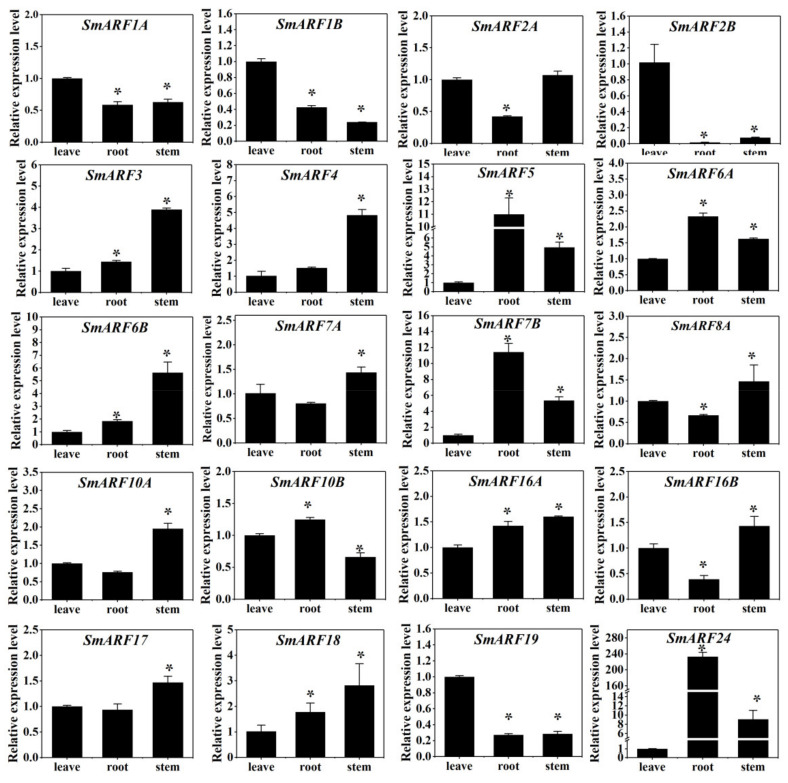
The expression profiles of *SmARF* genes in the leaves, roots, and stems of 3–week–old eggplants. The relative expression level was calculated using the method of 2^−^^△△Ct^. The relative mRNA levels of the leaves were used for the reference. The values are means ± SD (*n* = 3). * represents significance at *p* < 0.05 comparing with reference.

**Figure 7 ijms-23-06219-f007:**
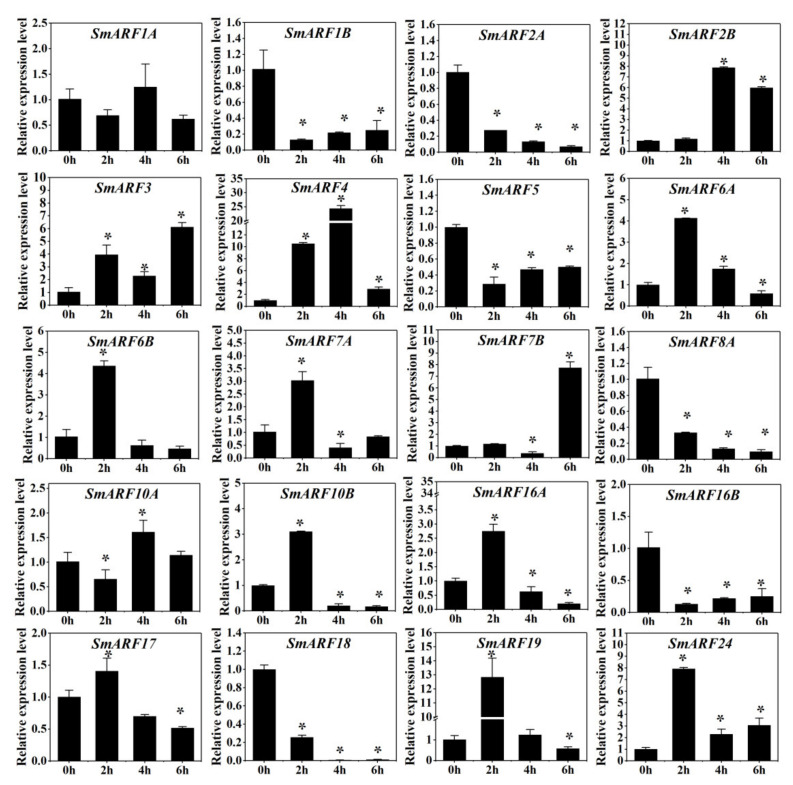
Expression analysis of *SmARFs* after 2,4–D treatment. The relative mRNA levels of the group at 0 h were used for the reference. The relative expression was calculated using the of 2^−^^△△Ct^ method. Values are means ± SD (*n* = 3). * represents significance at *p* < 0.05 compared with the reference.

**Figure 8 ijms-23-06219-f008:**
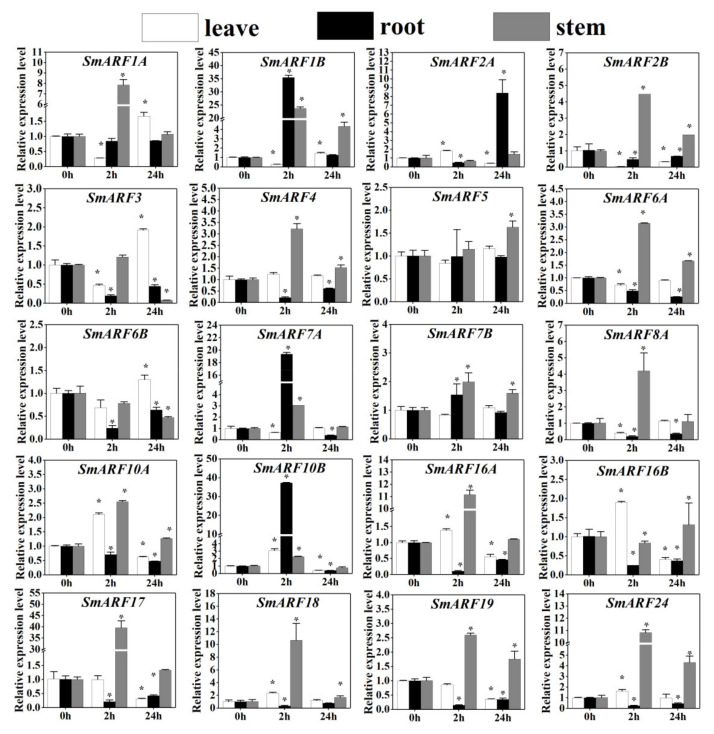
The expression profiles of auxin response factor (*ARF*) genes in eggplant (*Solanum melongena* L.) in response to NaCl treatment. The relative expression levels of 20 ARF genes in leaves, roots, and stems of 3 week–old eggplants were measured at 0, 2, and 24 h of treatment. The relative mRNA levels of the group at 0 h were used as for reference. The relative expression was calculated using the method of 2^−^^△△Ct^. Values are means ± SD (*n* = 3). * represents significance at *p* < 0.05 comparing with the references.

**Table 1 ijms-23-06219-t001:** Characteristics of putative genes encoding auxin response factors in *Solanum melongena*.

Gene Name	Gene ID	Amino Acids ^1^	PI ^2^	MW (kDa) ^3^	Domains	localization	Tomato Gene Name
SmARF1A	SMEL4.1_00g004780.1	766	6	85,257.51	Auxin_resp,B3,AUX_IAA	nucleus	SlARF1
SmARF1B	SMEL4.1_01g001560.1	766	6	85,257.51	Auxin_resp,B3,AUX_IAA	nucleus	SlARF1
SmARF2A	SMEL4.1_03g029420.1	869	6.7	96,739.8	Auxin_resp,B3,AUX_IAA	nucleus	SlARF2A
SmARF2B	SMEL4.1_05g012510.1	845	6.13	94,413.39	Auxin_resp,B3,AUX_IAA	nucleus	SlARF2B
SmARF5	SMEL4.1_04g022210.1	960	5.44	105,837.03	Auxin_resp,B3,AUX_IAA	nucleus	SlARF5
SmARF7B	SMEL4.1_05g024900.1	1153	5.92	126,924.92	Auxin_resp,B3, AUX_IAA	nucleus	SlARF7B
SmARF7A	SMEL4.1_07g007960.1	1168	6.24	129,081.62	Auxin_resp,B3,AUX_IAA	nucleus	SlARF7A
SmARF18	SMEL4.1_01g014290.1	748	5.86	83,734.67	Auxin_resp,B3,AUX_IAA(2)	nucleus	SlARF18
SmARF3	SMEL4.1_02g015730.1	807	6.75	87,259.21	Auxin_resp,B3	nucleus	SlARF3
SmARF4	SMEL4.1_12g016970.1	872	5.71	96,722.79	Auxin_resp,B3	nucleus	SlARF4
SmARF6A	SMEL4.1_12g018390.1	390	8.06	43,089.19	Auxin_resp,B3	nucleus	AtARF6
SmARF6B	SMEL4.1_07g013780.1	951	6.12	105,092.18	Auxin_resp,B3	nucleus	SlARF6
SmARF8A	SMEL4.1_03g001710.1	1052	6.23	116,913.79	Auxin_resp,B3	nucleus	SlARF8A
SmARF10A	SMEL4.1_12g017320.1	784	7.51	86,474.58	Auxin_resp,B3	nucleus	SlARF10A
SmARF10B	SMEL4.1_06g026750.1	737	6.8	82,152.63	Auxin_resp,WD40(2),B3	nucleus	SlARF10B
SmARF16A	SMEL4.1_09g002600.1	751	6.48	82,665.52	Auxin_resp,B3	nucleus	SlARF16A
SmARF16B	SMEL4.1_01g009500.1	669	6.08	74,853.8	Auxin_resp,B3	nucleus	SlARF16B
SmARF17	SMEL4.1_04g010090.1	407	7.72	44,721.66	Auxin_resp,B3	nucleus	SlARF17
SmARF24	SMEL4.1_08g000470.1	379	6.1	43,518.47	Auxin_resp,B3	nucleus	SlARF24
SmARF19	SMEL4.1_07g012550.1	1423	6.25	158,498.05	Auxin_resp, AUX_IAA	nucleus	SlARF19

^1^ Length of the amino acid sequence; ^2^ Molecular weight of the amino acid sequence; ^3^ Isoelectric point of the SmARF.

## Data Availability

Not applicable.

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
