# Peer review of "Genome–Wide Identification and Functional Characterization of Auxin Response Factor (ARF) Genes in Eggplant"

_ijms, 2022, doi:10.3390/ijms23116219_

Round 1
Reviewer 1 Report
The authors aim at identifying eggplant auxin response factor genes and carrying out an initial study on their tissue-specific expression patterns and responses to 2,4-D or salt treatments. The authors also carry out phylogenetic analyses to show the relationships between the identified SmARFs and ARFs from Arabidopsis and tomato. Although this is a promising study, it has not been carried out with enough attention to experimental set-up regarding the gene expression study. The authors are not showing data on non-treated plants after the hormonal treatments. Many Arabidopsis ARFs have circadian expression patterns, and therefore the observed changes in gene expression after the hormonal/salt treatments may be caused by circadian rhythmicity rather than the treatments. Without control samples, it is impossible to determine the cause of the observed changes. The lack of control samples is my main concern regarding this manuscript.
Specific comments:
The introduction provides justification for the current study by mentioning that auxin responses are important for increasing fruit set and salt stress tolerance in eggplant. However, it would be good to mention that eggplant is a close relative to tomato, a plant species in which the functions of ARFs have been studied to some extent. Given the close relationship of tomato and eggplant, the introduction should focus more on the known functions of SlARFs. This is much more relevant information than listing expression profiles of ARFs in very distantly related plant species, such as oil palm for example. The introduction should also give some more details on eggplant biology and the current release of the eggplant genome. For example, what is the ploidy level of eggplant? Is the current eggplant genome a draft version, or complete? This information is important for estimating the coverage of the identified eggplant ARFs. In the discussion, the authors mention on the lines #230-#231 "Additional genes might be uncovered in the regions of the eggplant genome that have not yet been sequenced". How well do the authors estimate to have covered eggplant ARFs?
The results are quite clearly presented, although figures need improvement. The font in figures is very small to unreadable. Some legends lack information.
In table 1, what is the eggplant genome version used?
Figure 1 shows an amino acid alignment of all the identified ARFs in the eggplant genome. I don't think that aligning aa sequences from a single species is very useful. It would be more interesting to see an alignment of eggplant ARFs against tomato ARFs (with shown functions). This would be a much more useful indication of protein functions.
In Fig4, what does the tree on the left side represent?
In Fig6, the legend should mention the statistical test used and the conditions under which these plant tissues were sampled.
The line #182 mentions "samples were treated in vitro.." What is the sample? Which tissue?
The discussion doesn't adequately compare the current results to earlier studies. For example, are the expression profiles of eggplant genes similar to those of closely related, phylogenetically close genes from tomato?
Materials and methods are not described with sufficient detail. For example, in 4.4 the source of the cloned SmARFs is not described. Which genotype
was used for cloning the cDNA? Which primers were used for cloning? If the genes were synthesized, by whom? What was the exact "promoter vector" used? What were the details of transiently expressing the SmARFs? How were the statistical analyses on gene expression data carried out and on which data? Note, that relative gene expression data is not normally distributed and therefore violates the assumptions of many statistical tests.
Author Response
Dear editor and reviewers:
Thank you for your letter and for the reviewers’ comments concerning our manuscript. Those comments are all valuable and very helpful for revising and improving our paper. We have modified the manuscript according to your kind advices and referee’s detailed suggestions which we hope meet with approval. Now I answer the questions one-by-one. Please see the attachment.
Reviewer 1:
Point 1: The introduction provides justification for the current study by mentioning that auxin responses are important for increasing fruit set and salt stress tolerance in eggplant. However, it would be good to mention that eggplant is a close relative to tomato, a plant species in which the functions of ARFs have been studied to some extent. Given the close relationship of tomato and eggplant, the introduction should focus more on the known functions of SlARFs. This is much more relevant information than listing expression profiles of ARFs in very distantly related plant species, such as oil palm for example. The introduction should also give some more details on eggplant biology and the current release of the eggplant genome. For example, what is the ploidy level of eggplant? Is the current eggplant genome a draft version, or complete? This information is important for estimating the coverage of the identified eggplant ARFs. In the discussion, the authors mention on the lines #230-#231 "Additional genes might be uncovered in the regions of the eggplant genome that have not yet been sequenced". How well do the authors estimate to have covered eggplant ARFs?
Response 1: Thanks for your suggestion. The following sentences have been added in the revised manuscript : “It has been reported that many SlARFs were differentially expressed in tomato leaves and roots under salt stress conditions and all SlARF genes were down-regulated in leaves except SlARF1, SlARF4, and SlARF19 whose expression was significantly induced [38]. SlARF8A and SlARF10A were significantly up-regulated by salt and drought stresses [38], while the down-regulation and loss function of SlARF4 increased tomato tolerance to salinity and drought stress [39]. In rice, both OsARF11 and OsARF15 showed differential expression in salt stress conditions [40]. Most BvARF genes in sugar beet were up-regulated or down-regulated to varying degrees by salt stress and with significant changes in nine genes [16]. In Tamarix chinensis, the expression profiling of TcARFs showed that only TcARF6 responds to salt stress [52].” (Line #62-#68 in revised manuscript).
The details on eggplant biology and the current release of the eggplant genome have been added as “The newest eggplant genome [41] obtained by scaffolding of v3.0 by the help of 3D chromosome conformation capture (Hi-C) information was used to identify the members of SmARF family using known SlARF protein sequences from tomato.” Line #71-#73 in revised manuscript.
In addition, the sentence: "Additional genes might be uncovered in the regions of the eggplant genome that have not yet been sequenced" has been deleted.
Point 2: The results are quite clearly presented, although figures need improvement. The font in figures is very small to unreadable. Some legends lack information.
Response 2: Thanks for your suggestion. We have improved figures quality and enlarge the font in figures.
Point 3: In table 1, what is the eggplant genome version used?
Response 3: Thanks for your suggestion. The reference has been added in the sentence as following: “…the eggplant genome reported by Barchi et al. (2021) [41] using BLASTP.” (Line #84-#85 in revised manuscript)
Point 4: Figure 1 shows an amino acid alignment of all the identified ARFs in the eggplant genome. I don't think that aligning aa sequences from a single species is very useful. It would be more interesting to see an alignment of eggplant ARFs against tomato ARFs (with shown functions). This would be a much more useful indication of protein functions.
Response 4: Thanks for your suggestion. Figure 1 has been changed to show an alignment of eggplant ARFs against tomato ARFs (with shown functions).
Point 5:In Fig4, what does the tree on the left side represent?
Response 5: We are sorry for our unclear description.Figure 4 has been revised as following:
Figure 4. Cis-acting regulatory elements upstream of auxin response factor genes in eggplant (Solanum melongena). The presence of cis-acting elements in a 2000-bp sequence upstream of the ATG start codons was assessed using PlantCARE. The horizontal scale indicates promoter length.
Point 6:In Fig6, the legend should mention the statistical test used and the conditions under which these plant tissues were sampled.
Response 6: Thanks for your suggestion. The legend of Figure 6 has been revised as: “The expression profiles of SmARF genes in the leaves, roots, and stems of 3-week-old eggplants. The relative expression level was calculated using the method of 2-△△Ct. Means were from three independent repeats; error bars indicate standard deviations from three biological replicates. Asterisks mark significant differences between organs and leave (t test). *P < 0.05 **P < 0.01 vs. the control.”
Point 7:The line #182 mentions "samples were treated in vitro.." What is the sample? Which tissue?
Response 7: Thanks for your suggestion. "samples were treated in vitro.." has been revised as “the 3 month-old eggplant flower buds before booming were treated with..” (Line #197 in revised manuscript)
Point 8: The discussion doesn't adequately compare the current results to earlier studies. For example, are the expression profiles of eggplant genes similar to those of closely related, phylogenetically close genes from tomato?
Response 8: Thanks for your suggestion. We have adequately compare the current results to earlier studies in the discussion.Some sentences have been revised or added as following: ‘’To identify the functions of SmARFs in eggplant, we analyzed their expression in leaves, roots, and stems. The data indicated that the expression of SmARFs was ubiquitous in all tissues. Among these genes, SmARF2B was significantly highly expressed in leaves. Similarly, the expressions of its tomato homolog, SlARF2B, were also significantly higher in leaves [25]. In Arabidopsis thaliana, AtARF2 could regulate leaf senescence [31]. Thus, we speculated that SmARF2B might play the similar role in eggplant leaves. However not all cases were the same. For example, SmARF10A was significantly high expressed in stems, while SlARF10A has been reported to be involved in the spatial restriction of the auxin response that drive leaf blade outgrowth in tomato [38]. Further work would be taken out to investigate the function of SmARFs on growth and development of eggplant.” (Line #261-#269 in revised manuscript).
“As transcription factor, ARFs participated in signal pathways related to auxin response and regulated the flower to fruit transition [38-40, 50-53]. The growth regulator 2, 4-D was used to mitigate flower abscission and increase fruit set. In this study, the expression of SmARF2B was significantly increased in flower buds after 2, 4-D treatment (Figure 7). The expression of its tomato homolog, SlARF2, was clearly responded to auxin. In addition, SlARF2 could regulate lateral root formation and flower organ senescence [47]. The silenced AtARF2/3/4 line leads to abnormal morphology of pollen grains [32]. The expression of SmARF7A was significantly increased after 2, 4-D treatment for 2 hours. The highly expression level of SlARF7 in placental tissues of the mature flower could activate auxin response attenuating genes that might repress the auxin response and prevent fruit set [45]. We also found SmARF10A was significantly increased after 2, 4-D treatment for 4 hours. It has been proved that SlARF10A, the target of sly-miR160, regulated auxin-mediated ovary patterning as well as floral organ abscission and lateral organ lamina outgrowth [48]. Thus, we speculated that SmARFs can exhibit many different functions in different plant species. ” (Line #272-#286 in revised manuscript).
“Transcription factors (TFs) coordinate gene expression by activating or inhibiting transcription in response to various abiotic stress signals [30, 38, 39-52]. In this study, the expression patterns of SmARF genes in response to NaCl treatment indicated that they might have crucial roles in eggplant leaves, roots, and stems (Figure 8). For example, the expression of SmARF2A was significantly decreased in 2h and increased in 24h after salt treatment. Previous studies have reported that the up-regulation of SlARF2 resulted in various asexual reproduction growth phenotypes, including increased lateral root formation [49]. Inhibition of AtARF2, AtARF3, and AtARF4 expression by tasiRNAs may release the repression of Arabidopsis lateral root growth [50]. In addition, SmARF7A was significantly repressed or induced respectively in leaves, roots, and stems under salt stress, which indicating that SmARF7A might be involved in stress response in eggplant. The expression profiles of SmARF7A were similar to SlARF7A which was phylogenetically close genes from tomato [38]. It had shown that AtARF7 is involved in the control of lateral root formation in Arabidopsis [30]. The other ARF transcription factors also played an essential role in the response to phytohormones and abiotic stress. For example, the IbARF5 gene from sweet potato (Ipomoea batatas) could increase tolerance to salt and drought stress in transgenic Arabidopsis [51]. In Tamarix chinensis, TcARF6 was rapidly expressed in response to salt stress but was significantly downregulated specifically in the roots [52].” (Line #272-#286 in revised manuscript)
Point 9: Materials and methods are not described with sufficient detail. For example, in 4.4 the source of the cloned SmARFs is not described. Which genotype
was used for cloning the cDNA? Which primers were used for cloning? If the genes were synthesized, by whom? What was the exact "promoter vector" used? What were the details of transiently expressing the SmARFs? How were the statistical analyses on gene expression data carried out and on which data? Note, that relative gene expression data is not normally distributed and therefore violates the assumptions of many statistical tests.
Response 9: Thanks for your suggestion. We have changed materials and methods 4.4 into ”The complete cds of ten SmARFs (SmARF1A, SmARF6A, SmARF2B, SmARF1B, SmARF18, SmARF8A, SmARF10B, SmARF24, SmARF16A, and SmARF10A) were obtained from the Eggplant Genome database [41]. HindIII and BamHI enzyme were used to linearize the CaMV35S promoter vector. PCR primers with 15-bp extensions (5’) that were complementary to the ends of the linearized vector were used to clone genes with PrimeSTAR Max DNA Polymerase (Takara, Dalian, China) from eggplant leaves. Sequences of 10 SmARFs were combined into CaMV35S promoter vectors with In-Fusion Snap Assembly cloning kits (Takara, Dalian, China) according to the manual. The recombinant vectors were then transformed into E. coli stains and the plasmids with correctly sequences were transformed into Agrobacterium GV3101. Use MMA suspension (1M MgCl2, 1M MES, 0.1M AS) to suspend the bacteria, make the concentration of OD600=0.6-0.8, adjust pH=5.6 and leave it at room temperature for 1-2h. The bacteriophage containing the target vector was mixed with p19 bacteriophage in equal volume, and the mixture was aspirated with a 1 mL syringe and injected into the abaxial surface of tobacco leaves. The protein localization was determined under a 20x confocal microscopy after 24 hours dark and 48 hours light. The primer sequences were listed in Table S1. "( Line #335-#353 in revised manuscript).
We have add “4.7.Statistical analysis: Data presented in this work are expressed as arithmetic means +/- SD of replicate plants within an experiment. The shown data are representative of a total of three independent biological replicates. The results were statistically analyzed using Dunnett’s t test; SPSS software version 23 (IBM, New York, USA). A P** <0.01 was considered statistically significant and a P* < 0.05 was considered statistically. In all the presented figures, error bars indicate standard deviation.”(line#354-#359 in revised manuscript)

Reviewer 2 Report
Chen et al. reported the genome-wide identification and functional characterization of auxin response factor genes in eggplant. This work is fairly interesting and might attract the readership of the International Journal of Molecular Sciences. However, the current manuscript suffers from several issues that need to be addressed to improve its quality.
- Most of the figures in the manuscript are of low quality and the accompanying wordings are illegible. For example, most of the wordings in Figure 1 are hardly readable. The same problem exists for Figure 3.
- The authors stated that all ten fusion proteins were detected in the nucleus. This statement is not convincing. The quality of Figure 5 is very poor, and it is extremely hard to spot the nucleus. The authors need to improve the image quality. Furthermore, the authors are suggested to explore other characterization, in addition to confocal microscopy, to show the localization of the proteins in nucleus.
- Figures 6 to 8 have poor quality. The authors need to improve their quality and enlarge the figure wordings.
- There are some errors in the numbering of the sections. For example, on page 11, sections 2.5 and 2.6 should be 4.5 and 4.6, respectively.
- The authors failed to provide the methods they used for statistical analysis. A detailed section on statistical analysis needs to be included.
Author Response
Dear editor and reviewers:
Thank you for your letter and for the reviewers’ comments concerning our manuscript. Those comments are all valuable and very helpful for revising and improving our paper. We have modified the manuscript according to your kind advices and referee’s detailed suggestions which we hope meet with approval. Now I answer the questions one-by-one. Please see the attachment.
Point 1: Most of the figures in the manuscript are of low quality and the accompanying wordings are illegible. For example, most of the wordings in Figure 1 are hardly readable. The same problem exists for Figure 3.
Response 1: Thanks for your suggestion. We are sorry for the low quality figures in the previous manuscript. Figure 1 and 3 have been revised in our rexision. Please see the attachment.
Point 2: The authors stated that all ten fusion proteins were detected in the nucleus. This statement is not convincing. The quality of Figure 5 is very poor, and it is extremely hard to spot the nucleus. The authors need to improve the image quality. Furthermore, the authors are suggested to explore other characterization, in addition to confocal microscopy, to show the localization of the proteins in nucleus.
Response 2: Thanks for your suggestion. Figure 5 has been revised in our revision. Please see the attachment.
Point 3: Figures 6 to 8 have poor quality. The authors need to improve their quality and enlarge the figure wordings.
Response 3: Thanks for your suggestion. We are so sorry for providing poor quality figures that we have improved their quality and enlarged the figure wordings. Figure 6-8 have been revised in our revision. Please see the attachment.
Point 4: There are some errors in the numbering of the sections. For example, on page 11, sections 2.5 and 2.6 should be 4.5 and 4.6, respectively.
Response 4: Thanks for your suggestion. We are so sorry for making errors in the numbering of sections that we have corrected the errors carefully.
Point 5: The authors failed to provide the methods they used for statistical analysis. A detailed section on statistical analysis needs to be included.
Response 5: Thanks for your suggestion. We have added statistical analysis in materials and methods 4.7 “Data presented in this work were expressed as arithmetic means +/- SD of replicate plants within an experiment. The shown data were representative of a total of three independent biological replicates. The results were statistically analyzed using Dunnett’s t test; SPSS soflware version 23 (IBM, New York, USA). A P** < 0.01 was considered statistically significant and a P* < 0.05 was considered statistically. In all the presented figures, error bars indicated standard deviation.”(lines#354-#359 ).

Round 2
Reviewer 1 Report
The authors have addressed most of my comments sufficiently well. However, the figures and figure legends could be still improved. For example, Figure 1 shows an alignment of different protein domains. The aa headings on the left suggest that tomato sequence are included and there is also some sort of color code used to distinguish eggplant and tomato sequences. However, this is not mentioned in the legend. Please improve the legends.
Figure 3 font is still nearly impossible to read and the figure resolution is poor.
Figure 6 font is too small. In this figure, I'm still not convinced of the statistics. Relative expression values as presented here are not normally distributed and I have serious doubts about the homogeneity of variance as well (=assumptions for t-test are not met). Violating the assumptions of the t-test may lead to incorrect P-values. Please improve the statistics here. The same comments apply to Figures 7 and 8.
Some descriptions are still vague in the Mat&met chapter. For example, the transient transformation of tobacco leaves is apparently done using agrobacterium. But what is the bacteriophage mentioned here? To my knowledge, it is not common to use a bacterial virus in conjunction with agrobacterium. Please clarify. Also, make a careful revision of English used in the mat&met; incorrect spelling or grammar makes the descriptions difficult to understand at points. Which eggplant genotype/cultivar was used in the study?
Author Response
Dear editor and reviewers:
Thank you for your letter and for the reviewers’ comments concerning our manuscript. Those comments are all valuable and very helpful for revising and improving our paper. We have modified the manuscript according to your kind advices and referee’s detailed suggestions which we hope meet with approval. Now I answer the questions one-by-one.
Point 1: The authors have addressed most of my comments sufficiently well. However, the figures and figure legends could be still improved. For example, Figure 1 shows an alignment of different protein domains. The aa headings on the left suggest that tomato sequence are included and there is also some sort of color code used to distinguish eggplant and tomato sequences. However, this is not mentioned in the legend. Please improve the legends.
Response 1: Thanks for your suggestion. We have improved the legends as following :”Multiple alignment profiles of B3 (DNA binding domain), auxin_resp, and CTD (Aux/IAA-binding) domains of SmARF proteins obtained with the Jalview program. Red and blue code used to distinguish eggplant and tomato sequences. All sequences show high level of amino acids conservationin blue. The logo at the top shows amino acids, conservation in large letter represent.”.
Point 2: Figure 3 font is still nearly impossible to read and the figure resolution is poor.
Response 2: Thanks for your suggestion. Figure 3 has been splitted into two part including revised Figure 3 and Figure S1 .Please see the attachment.
Point 3: Figure 6 font is too small. In this figure, I'm still not convinced of the statistics. Relative expression values as presented here are not normally distributed and I have serious doubts about the homogeneity of variance as well (=assumptions for t-test are not met). Violating the assumptions of the t-test may lead to incorrect P-values. Please improve the statistics here. The same comments apply to Figures 7 and 8.
Response 3: Thanks for your suggestion. We have improved Figure 6、7 and 8. Please see the attachment.
Point 4: Some descriptions are still vague in the Mat&met chapter. For example, the transient transformation of tobacco leaves is apparently done using agrobacterium. But what is the bacteriophage mentioned here? To my knowledge, it is not common to use a bacterial virus in conjunction with agrobacterium. Please clarify. Also, make a careful revision of English used in the mat&met; incorrect spelling or grammar makes the descriptions difficult to understand at points. Which eggplant genotype/cultivar was used in the study?
Response 4: Thanks for your suggestion. We have improved the mat&met as following:”The complete cds of ten SmARFs (SmARF1A, SmARF6A, SmARF2B, SmARF1B, SmARF18, SmARF8A, SmARF10B, SmARF24, SmARF16A, and SmARF10A) were obtained from the Eggplant Genome database [41]. HindIII and BamHI enzyme were used to linearize the CaMV35S promoter vector with GFP. Sequences of 10 SmARFs were combined into the vector using In-Fusion Snap Assembly cloning kits (Takara, Dalian, China) according to the manual. The fusion construct was transferred into Agrobacterium tumefaciens strain GV3101, and subcellular localization assays were performed as previously reported [62]. The protein localization was determined under a 20X confocal microscopy. The primer sequences were listed in Table S1.”(Line #92-#103 in revised manuscript).
“The high generation inbred line No.108 [63] seeds were sterilized for 10 min in 50% sodium hypochlorite, rinsed four times with sterile distilled water, and sown in pots containing peat.” (Line #105-#107 in revised manuscript).

Reviewer 2 Report
The authors have not addressed most of my comments satisfactorily. The quality of most of the figures is still very poor, and I am not able to evaluate the merit of the manuscript appropriately.
Author Response
Dear editor and reviewers:
Thank you for your letter and for the reviewers’ comments concerning our manuscript. Those comments are all valuable and very helpful for revising and improving our paper. We have modified the manuscript according to your kind advices and referee’s detailed suggestions which we hope meet with approval. Now I answer the questions one-by-one.
Point 1: The authors have not addressed most of my comments satisfactorily. The quality of most of the figures is still very poor, and I am not able to evaluate the merit of the manuscript appropriately.
Response 1: Thanks for your suggestion. We are sorry for the low quality figures in the previous manuscript. We have improved the Figures. Please see the attachment.

Round 3
Reviewer 2 Report
The manuscript quality has been improved.